# TAPAS: Datasets for Learning the Learning with Errors Problem

**Eshika Saxena**[*]
FAIR at Meta

**Alberto Alfarano**
FAIR at Meta

**François Charton**[†]
FAIR at Meta

**Emily Wenger**[†]
Duke University

**Kristin Lauter**[†]
FAIR at Meta

## Abstract

AI-powered attacks on Learning with Errors (LWE)—an important hard math problem in post-quantum cryptography—rival or outperform "classical" attacks on LWE under certain parameter settings. Despite the promise of this approach, a dearth of accessible data limits AI practitioners' ability to study and improve these attacks. Creating LWE data for AI model training is time- and compute-intensive and requires significant domain expertise. To fill this gap and accelerate AI research on LWE attacks, we propose the TAPAS datasets, a **t**oolkit for **a**nalysis of **p**ost-quantum cryptography using **AI s**ystems. These datasets cover several LWE settings and can be used off-the-shelf by AI practitioners to prototype new approaches to cracking LWE. This work documents TAPAS dataset creation, establishes attack performance baselines, and lays out directions for future work.

## 1 Introduction

The Learning with Errors (LWE) problem is a hard math problem used in post-quantum cryptography. Put simply, the LWE problem is: given a set of samples $(\mathbf{a}, b) \in \mathbb{Z}_q$ where $b$ is the noisy inner product of $\mathbf{a}$ with a secret vector $\mathbf{s}$, e.g. $b = \mathbf{a} \cdot \mathbf{s} + \mathbf{e} \in \mathbb{Z}_q$, recover $\mathbf{s}$. LWE is attractive for use in post-quantum cryptography because no classical **or** quantum algorithms are known to recover $\mathbf{s}$ given many $(\mathbf{a}, b)$ samples in polynomial time under certain conditions. Thus, LWE can serve as an alternative to current hard math problems like integer factorization used in cryptosystems like RSA, which can be broken by quantum computers due to Shor's algorithm [33]. Consequently, LWE is used in several US-government standardized PQC systems, such as CRYSTALS-KYBER [4], and also in homomorphic encryption applications [2], which enable computation on encrypted data.

Given the importance of LWE in securing the future internet, understanding its security is critical. Cryptanalysis of LWE and LWE-based cryptosystems is a long-studied problem in theoretical cryptography, and analysis from this community has not uncovered any notable weaknesses of LWE. However, vigorous security analysis demands examining LWE from multiple angles, including using previously-underutilized tools like AI, to ensure it is secure against a range of attacks.

**AI attacks on LWE match or outperform classical attacks in certain settings.** A series of recent works [35, 21, 20, 34] has explored a novel approach using AI models to recover LWE secrets. The attack setup is simple: train a model using millions of LWE samples $(\mathbf{a}, b)$ (generated from a small initial sample set) to predict $b$ given a particular $\mathbf{a}$. If the model performs better than random on this task, it must have implicitly learned information about the secret $\mathbf{s}$. The secret can then be recovered via carefully crafted model queries.

---

[*]Corresponding author: eshika@meta.com
[†]Co-senior authors

39th Conference on Neural Information Processing Systems (NeurIPS 2025) Track on Datasets and Benchmarks.

Traditional LWE cryptanalysis methods treat lattice-based systems as a mathematical problem to be solved rather than data to be analyzed. The AI approach inverts this paradigm, instead searching for patterns in large quantities of LWE data. Not only is this AI approach novel, it is effective: AI models match or outperform existing "classical" attacks on lattice cryptography under certain conditions [36]. Additional work [23] suggests that further helpful cryptanalytic insights could emerge from enhancing this data-centric approach to LWE. Finally, recent work argues that such AI-based cryptanalysis approaches are not inherently limited, raising the possibility that they could scale further [32].

**AI research on LWE (and cryptography) benefits both fields.** The intersection of AI and cryptography is a long-recognized [28] but relatively unexplored area, offering many opportunities for meaningful research. By studying the application of AI techniques to LWE cryptanalysis, researchers can unlock new techniques and approaches to complement those of [35, 20, 21, 34]. Both the AI and cryptography communities would benefit from this.

On the cryptography side, additional work on the AI approach may lead to the discovery of better attacks on LWE. Discovering and mitigating these before LWE is widely deployed is critical for internet security. The development of more robust cryptographic protocols will have important implications for secure communication and data protection.

On the AI side, studying AI attacks on LWE will require AI researchers to tackle critical problems for enabling higher-order mathematical reasoning in AI models. Currently, AI models struggle to perform mathematical operations such as modular arithmetic, which is a fundamental part of the LWE problem [16, 24]. As AI practitioners gain exposure to important cryptography problems, further investigation into how and why AI models fail on these problems will help advance the capabilities of AI models.

**Dataset availability roadblocks AI LWE research.** Despite these myriad benefits to both communities, it remains difficult for AI researchers to begin studying LWE attacks. Running AI attacks on LWE requires significant *preprocessing* of LWE samples, which enables models to learn better [21]. Preprocessing can take hours to days, can be computationally expensive, and requires some expertise in lattice reduction algorithms—all of which can deter AI practitioners who may lack time, compute, and domain knowledge. Prior work has open-sourced some LWE datasets [36, 20], but these contain only a few million LWE samples, making it difficult to train bigger models or develop scaling laws. Additionally, published datasets only consider difficult LWE parameter regimes that may not prove accessible (computationally or otherwise) for AI researchers.

**Our Contribution: large-scale, open-source LWE datasets.** For the AI community to contribute meaningfully to the study of LWE cryptanalysis, large datasets of reduced LWE samples must exist and be publicly accessible. Such datasets would provide the AI community with an easy entry point to the important and often inaccessible research intersection between AI and cryptography. This motivates our work: *providing large-scale datasets of preprocessed Learning with Errors (LWE) data across many parameter settings* to make this research more accessible to the AI community. To this end, this paper provides the following:

- **Five new datasets of LWE samples**, designed for off-the-shelf use in AI applications, hosted on Huggingface[1] for easy access.
- **Baseline results for SALSA and Cool and Cruel attack performance on provided datasets**, setting a baseline on which future work can improve.

**Paper organization.** The rest of the paper proceeds as follows. §2 gives context on the Learning with Errors problem and prior cryptanalytic approaches. §3 describes the datasets we provide. §4 gives baseline attack performance on datasets. §5 discusses related datasets and benchmarks, and §7 suggests ways our datasets can be used to enhance AI attacks on LWE.

## 2 Background: Cryptanalysis of Learning with Errors

The Learning with Errors (LWE) problem was first proposed for cryptography by Regev [26, 27]. Since no classical or quantum algorithms are known to solve LWE in polynomial time for certain parameter settings, it has been widely adopted for use in post-quantum cryptosystems. Substantial

---

[1]https://huggingface.co/datasets/facebook/TAPAS

efforts have been made to discover weaknesses in LWE, to ensure it is a solid foundation for post-quantum cryptography. This section describes the basics of LWE and gives an overview of attacks against it.

**LWE Basics.** There are two variants of the LWE problem: Search-LWE and Decision-LWE. The Search-LWE problem is: given samples $(\mathbf{A}, \mathbf{b})$—where $\mathbf{A} \in \mathbb{Z}_q^{m \times n}$ is populated with uniform random entries modulo a large prime $q$, and $\mathbf{b} = \mathbf{A} \cdot \mathbf{s} + \mathbf{e} \in \mathbb{Z}_q^m$ is the noisy matrix-vector product of $\mathbf{A}$ and a secret vector $\mathbf{s} \in \mathbb{Z}_q^n$ with error vector $\mathbf{e}$—recover the secret vector $\mathbf{s}$. In Decision-LWE, one is tasked with deciding whether $(\mathbf{A}, \mathbf{b})$ are LWE samples or were generated uniformly at random. The secret $\mathbf{s}$ and error $\mathbf{e}$ are chosen from some probability distributions, which we denote by $\chi_s$ and $\chi_e$. The hardness of LWE depends on $n$, $q$, and these distributions. We denote a single row of $(\mathbf{A}, \mathbf{b^t})$ as $(\mathbf{a}, b) \in \mathbb{Z}_q^n \times \mathbb{Z}_q$, and use this notation going forward to refer to a specific LWE sample.

**"Classical" Attacks on LWE.** Generally, classical or algebraic attacks on LWE try to find short vectors in a lattice $\Lambda$ constructed from samples $(\mathbf{A}, \mathbf{b})$. Finding a short enough vector is sufficient to recover the LWE secret [25]. Most approaches leverage lattice reduction algorithms like LLL and BKZ [19, 31, 10] to *reduce* or shorten the vectors in $\Lambda$. They then run additional algorithms to recover $\mathbf{s}$ from this reduced lattice basis, e.g. [11, 13, 37, 1, 5, 8, 22, 7, 3]. See [36] for a broad overview of classical attacks on LWE.

**AI Attacks on LWE.** So far, two AI-powered attacks have been proposed against LWE: SALSA and its variants [35, 20, 21, 34] and Cool & Cruel [23]. The SALSA attack works as follows. Start with a set of $4n$ LWE pairs $(\mathbf{a}, b)$, where $n$ is the length of the vector $\mathbf{a}$. Through repeated sub-sampling of these $4n$ samples, create several million new LWE matrices $\mathbf{A} \in \mathbb{Z}_q^{m \times n}$ with corresponding $\mathbf{b} = \mathbf{A} \cdot \mathbf{s} + \mathbf{e}$. Run lattice reduction on these samples (further details in §3) to produce reduced samples $(\mathbf{RA}, \mathbf{Rb} = \mathbf{RA} \cdot \mathbf{s} + \mathbf{Re})$. Then, train a model $\mathcal{M}$ to predict $\mathbf{Rb}$ from $\mathbf{RA}$. If this model ever achieves better-than-random performance on this task, it has implicitly learned $\mathbf{s}$, and can be selectively queried to recover $\mathbf{s}$.

The Cool & Cruel attack takes a different but related approach. It uses the same subsampling-then-reduce procedure from SALSA to produce $(\mathbf{RA}, \mathbf{Rb})$ pairs. However, it then observes an asymmetry in the columns of $\mathbf{RA}$ due to a quirk of the lattice reduction algorithms—the first columns of $\mathbf{RA}$ are not reduced, while the last columns are. This asymmetry admits an attack in which the entries of the secret $\mathbf{s}$ corresponding to the unreduced $\mathbf{RA}$ columns can be guessed via brute force, while the other entries of $\mathbf{s}$ can be recovered via linear regression (making this an "AI" attack) [36].

**Comparing Classical and AI Attacks on LWE.** Prior work [36] provided the first empirical comparisons of classical and AI attacks on LWE and found AI attacks matched or outperformed classical attacks on certain weakened LWE settings (realistic $n$ and $q$ sizes, sparse secrets). This work motivates our own, as it demonstrates that AI attacks are already competitive in this space.

# 3 TAPAS: Datasets for Learning the Learning with Errors Problem

Given the competitive performance of AI attacks on LWE and the ongoing need to assess the security of LWE-based cryptosystems, our goal is to make LWE attack development accessible to AI researchers. To this end, we provide five datasets that are ready for AI model training off the shelf. Building on the naming convention of the initial AI attacks on LWE (SALSA, SALSA Picante, etc), we refer to the provided datasets collectively as TAPAS: a **T**oolkit for **A**nalysis of **P**ost-quantum cryptography using **AI S**ystems. These datasets cover a wide range of LWE hardness settings, as controlled by the lattice dimension $n$, modulus $q$, and secret/error distributions $\chi_s$ and $\chi_e$. By providing a variety of datasets with varying hardness, we hope to enable AI researchers to prototype novel approaches on scaled-down versions of the LWE problem and then also scale up to larger, more realistic datasets. This section presents details of our LWE datasets, including their parameters and development process.

## 3.1 Dataset Overview

**Important LWE parameters.** We first give an overview of important parameters for LWE, which are relevant for attacks on LWE. Every instance of LWE is defined by a lattice dimension $n$, a modulus

$q$, and secret and error distributions $\chi_s$ and $\chi_e$, which determine problem hardness. We also briefly discuss variants of LWE like R-LWE and MLWE, although these are not used in our datasets.

- **Dimension** $n$ is the number of entries in the random vector $\mathbf{a}$. In practical LWE implementations, $n$ is a power of 2, to avoid powerful attacks on LWE variants when $n$ is not a power of 2 [14].
- The **prime modulus** $q$ defines the field where all lattice operations are carried out. All vector and lattice entries are integers modulo $q$.
- The coordinates of the secret $\mathbf{s}$ are chosen from a **secret distribution** $\chi_s$. Although early implementations of LWE chose secret coordinates uniformly at random from $\mathbb{Z}_q$, computational efficiency and improved functionality motivates choosing small entries for $\mathbf{s}$. So secrets are often chosen from narrow distributions such as binary and ternary secrets, $\mathbf{s} \in \{0, 1\}$ or $\mathbf{s} \in \{-1, 0, 1\}$, recommended for use in homomorphic encryption operations [2, 6], where efficiency is key. Similarly, in the standardized CRYSTALS-KYBER system, $\chi_s$ is a binomial distribution with $\eta = 3$, which corresponds to $\mathbf{s} \in \{-3, -2, -1, 0, 1, 2, 3\}$.
- The coordinates of the error $\mathbf{e}$ are chosen from an **error distribution** $\chi_e$. For homomorphic encryption applications $\chi_e = N(0, 3)$ (rounded to the nearest integer) [2, 6], regardless of the lattice dimension or modulus size. For CRYSTALS-KYBER, $\chi_e$ is again binomial with $\eta = 3$, so $\mathbf{e} \in \{-3, -2, -1, 0, 1, 2, 3\}$.
- LWE has several **variants**. In basic LWE, the coordinates of $\mathbf{A}$ are chosen uniformly at random from $\mathbb{Z}_q$. However, this approach is computationally inefficient, since it requires storing a full $n \times n$ matrix. To address this, Ring-LWE was proposed, in which a sample is defined as $(a(x), b(x) = a(x)s(x) + e(x))$ here $a(x), b(x)$ are polynomials in a cyclotomic ring $R_q = \mathbb{Z}_q[X]/(X^n + 1)$ and $n$ is a power of 2. RLWE is widely used in homomorphic encryption applications [2, 6] and only requires storing the n-long polynomial vector. Yet another LWE variant is Module Learning with Errors (MLWE), which builds on RLWE but works in a free $R_q$-Module $\mathcal{M} = R_q^k$ of rank $k$. An MLWE sample is a pair $(\mathbf{a}, b)$ where $\mathbf{a} = (a_1(x), a_2(x), \ldots, a_k(x)) \in \mathcal{M}$, and $b = \mathbf{a} \cdot \mathbf{s} + e \in R_q$ for some secret vector of polynomials $\mathbf{s} = (s_1(x), s_2(x), \ldots, s_k(x)) \in \mathcal{M}$, and error polynomial $e$ chosen from a specified distribution. MLWE is used in CRYSTALS-KYBER [4] and is between LWE and RLWE in terms of computational efficiency, requiring storage of $k$ $n$-long vectors. Although RLWE and MLWE are important, this work considers only LWE for simplicity.

**Our datasets.** In this work, we release 5 reduced LWE datasets with varying parameter settings. Table 1 gives an overview of the parameters for these datasets. The parameters are chosen to offer a wide range of problem hardness, including several parameter settings used in standardized (or proposed standardized) LWE-based cryptosystems. Access to datasets with varying parameter settings enables investigation along many different axes to see what affects attack performance, such as: $n$ (sequence length), $q$ (size of the modulus), and $\rho$ (quality of reduction). In this work, we provide datasets with 10x (40 million) and 100x (400 million) the number of samples from prior work to enable further research on how the number of samples affects attack performance.

Table 1: **Overview of datasets provided in this work**. $n$ and $q$ are the LWE dimension and modulus, respectively. $\omega$ is the penalty parameter used in reduction (prior experimental work found that 10 is sufficient for Gaussian error with $\sigma = 3.2$ [21]). $\rho$ is the stddev reduction, a metric reported in [20].

| $n$ | $\log_2 q$ | $\omega$ | $\rho$ | # samples |
|------|------------|----------|--------|-----------|
| 256 | 20 | 10 | 0.4284 | 400M |
| 512 | 12 | 10 | 0.9036 | 40M |
| 512 | 28 | 10 | 0.6740 | 40M |
| 512 | 41 | 10 | 0.3992 | 40M |
| 1024 | 26 | 10 | 0.8600 | 40M |

## 3.2 Data Generation

To create our datasets, we leverage the data preprocessing techniques first proposed in [21] and improved in [23]. All prior work on applying AI to LWE problems found that this preprocessing step was critical to improving AI performance on the task, so we believe it is advantageous to the community to publish preprocessed datasets.

Each dataset starts with a set of $4n$ LWE samples $(\mathbf{A}, \mathbf{b}) \in \mathbb{Z}_q^{4n \times n}, \mathbb{Z}_q^{4n}$. We assume the samples are eavesdropped, a common assumption in LWE attack literature. Then, we employ the subsampling trick of [21] to create millions of new LWE samples from these: select $m$ random indices from the $4n$ set to form $(\mathbf{A}_i, \mathbf{b}_i) \in \mathbb{Z}_q^{m \times n}, \mathbb{Z}_q^m$. This trick allows us to create up to $\binom{4n}{m}$ samples from the initial starting set—billions more samples than we will actually need.

To "preprocess" this data and create the training datasets, we then create a q-ary lattice embedding $\mathbf{\Lambda}_i$ of each subsampled $\mathbf{A}_i$ via:

$$\mathbf{\Lambda}_i = \begin{bmatrix} 0 & q \cdot \mathbf{I}_n \\ \omega \cdot \mathbf{I}_m & \mathbf{A}_i \end{bmatrix} \tag{1}$$

Lattice reduction on $\mathbf{\Lambda}_i$ finds a unimodular transformation $[\mathbf{L} \quad \mathbf{R}]$ which minimizes the norms of $[\mathbf{L} \quad \mathbf{R}] \mathbf{\Lambda_i} = [\omega \cdot \mathbf{R} \quad \mathbf{RA} + q \cdot \mathbf{L}]$. $\omega$ is a scaling parameter that trades-off reduction strength and the error introduced by reduction. This $\mathbf{R}$ matrix is then applied to the original $(\mathbf{A}_i, \mathbf{b}_i)$ to produce reduced samples $(\mathbf{RA}_i, \mathbf{Rb}_i)$ with smaller norms. Repeating this process many times (parallelized across many CPUs) produces a dataset of reduced LWE samples.

To run lattice reduction on $\mathbf{\Lambda}_i$, we interleave two popular reduction algorithms, BKZ2.0 [10] and flatter [29], following the approach of [23]. After each reduction algorithm completes a step, we run the polishing algorithm of [9]. These algorithms are parameterized by a blocksize $\beta$ (for BKZ2.0); a reduction strength parameter $\alpha$ (for flatter); algorithm switching parameters $\gamma$ and $s$, which represent the minimum amount ($\gamma$) by which the reduction must improve over $s$ steps of a particular algorithm, otherwise an algorithm switch is triggered; and a data writing threshold $\tau$. An overview of our reduction approach, which describes the purpose of these parameters, is given in Algorithm 1 and 2.

---

**Algorithm 1** Interleaved lattice reduction

---

InterleavedReduction($\Lambda_i, \alpha, \beta, \gamma, s, \tau$):

$\rho = \inf$; {set reduction threshold at infinity}
prior_$\rho$ = [];
algo1 = Flatter($\alpha$) [29]; $a_1$ = True; {flatter goes first}
algo2 = BKZ2.0($\beta$) [10]; $a_2$ = False; {BKZ2.0 goes second}
**while** $\rho \geq \tau$ **do**
  **while** $a_1$ **do**
    $\Lambda_i$ = polish(algo1($\Lambda_i$));
    $a_1, a_2, \rho$, prior_$\rho$ = CheckForSwitch($\Lambda_i$, $\rho$, prior_$\rho$, s, $\gamma$, $a_1$, $a_2$);
    **if** $\rho \leq \tau$ **then**
      break
    **end if**
  **end while**
  **while** $a_2$ **do**
    $\Lambda_i$ = polish(algo2($\Lambda_i$));
    $a_1, a_2, \rho$, prior_$\rho$ = CheckForSwitch($\Lambda_i$, $\rho$, prior_$\rho$, s, $\gamma$, $a_1$, $a_2$);
    **if** $\rho \leq \tau$ **then**
      break
    **end if**
  **end while**
**end while**
return $\Lambda_i$;

---

**Algorithm 2** Criteria for switching reduction algorithms based on reduction progress.

---

CheckForSwitch($\Lambda_i, \rho$, prior_$\rho, s, \gamma, a_1, a_2$):

stall = False; {Assume we aren't stuck.}
**if** len(prior_$\rho$) $> s + 1$ **then**
  decreases = [prior_$\rho[i - 1]$ - prior_$\rho[i]$ for $i$ in range$(-s, 0)$]
  **if** mean(decreases) $< \gamma$ **then**
    stall = True;
  **end if**
**end if**
**if** stall **then**
  $a_1$ =!a1
  $a_2$ =!a2
**end if**
prior_$\rho$.append($\rho$);
$\rho$ = ComputeReduction($\Lambda_i$);
return $a_1, a_2, \rho$, prior_$\rho$;

---

### 3.3 Implementation Details

Lattice reduction is implemented in Python, utilizing the `fpylll` and `flatter` libraries for the BKZ2.0 and flatter implementations[2]. We run lattice reduction on CPUs only (2.1GHz Intel Xeon Gold CPUs with 750 GB of RAM). Table 2 specifies the reduction parameter values used. These were selected after substantial empirical evaluation of different parameter values. We found in general that, for each matrix, the amount of reduction tended to "flatline" after an initial period of significant improvement (see Figure 1), so we optimized parameters to achieve maximal reduction in reasonable time before performance flatlined.

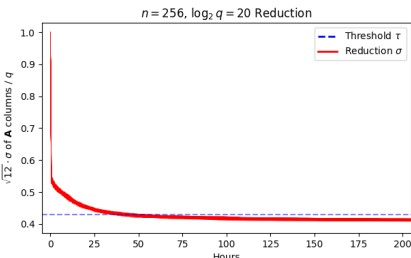 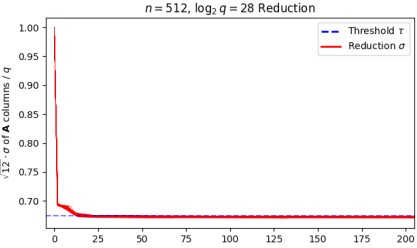

Figure 1: **Reduction over time for** $N = 256, \log q = 20$ **(left) and** $N = 512, \log q = 28$ **(right).** *Threshold $\tau$ denoted by the dashed blue line. Each red line denotes a separate reduction experiment.*

Table 2: **Reduction parameters.** *The writing threshold $\tau$ is customized for each $n/q$ setting based on extensive experiments. Most other parameters are fixed across all reduction experiments.*

| Parameter | $\alpha$ | $\beta$ | $\gamma$ | $s$ | $\tau$ | $\omega$ |
|---|---|---|---|---|---|---|
| **Setting** | $0.04$ | $30$ if $n \le 256$ else $18$ | $-0.001$ | $3$ | Varies by $n/q$ | $10$ |

## 4 Baseline Results

### 4.1 Data Quality Analysis

We present different statistics for each of the datasets in Table 3. We measure the reduction factor $\rho$ as the ratio of the means of the standard deviations of the rows of $\mathbf{RA}$ and $\mathbf{A}$. A smaller $\rho$ implies that $\mathbf{RA}$ is more reduced. We preprocess each matrix $\mathbf{A}$ until $\rho$ stops decreasing. For each dataset, we report $\rho$ to 4 decimal places. In addition, per [23], the reduction algorithms produce a cliff shape in the standard deviations of the columns of $\mathbf{RA}$ (see Figure 2). Columns with standard deviations greater than $\frac{q}{\sqrt{12}}$ form the "cruel" region in the cliff because secret bits in those column indices are more difficult to recover. In Table 3, we also report the size of the cliff produced by the lattice reduction algorithms.

Table 3: **Statistics on the datasets provided by this work**. *$n$ and $q$ are the LWE dimension and modulus, respectively. $m$ is the number of random indices we select at a time for subsampling. $\rho$ is the stddev reduction, a metric reported in [20], while # cruel bits is the number of unreduced bits, used by [23]. # samples/matrix is slightly less than $m + n$ as we remove the rows with all zeroes. All statistics are calculated on a representative subset of 5000 examples.*

| $n$ | $\log_2 q$ | $m$ | $\rho$ | # cruel bits | # samples/matrix |
|---|---|---|---|---|---|
| 256 | 20 | 224 | 0.4284 | 37 | 438 |
| 512 | 12 | 448 | 0.9036 | 411 | 841 |
| 512 | 28 | 448 | 0.6740 | 225 | 939 |
| 512 | 41 | 448 | 0.3992 | 69 | 938 |
| 1024 | 26 | 1624 | 0.8600 | 750 | 2626 |

---

[2]https://github.com/facebookresearch/LWE-benchmarking/tree/main/src/generate

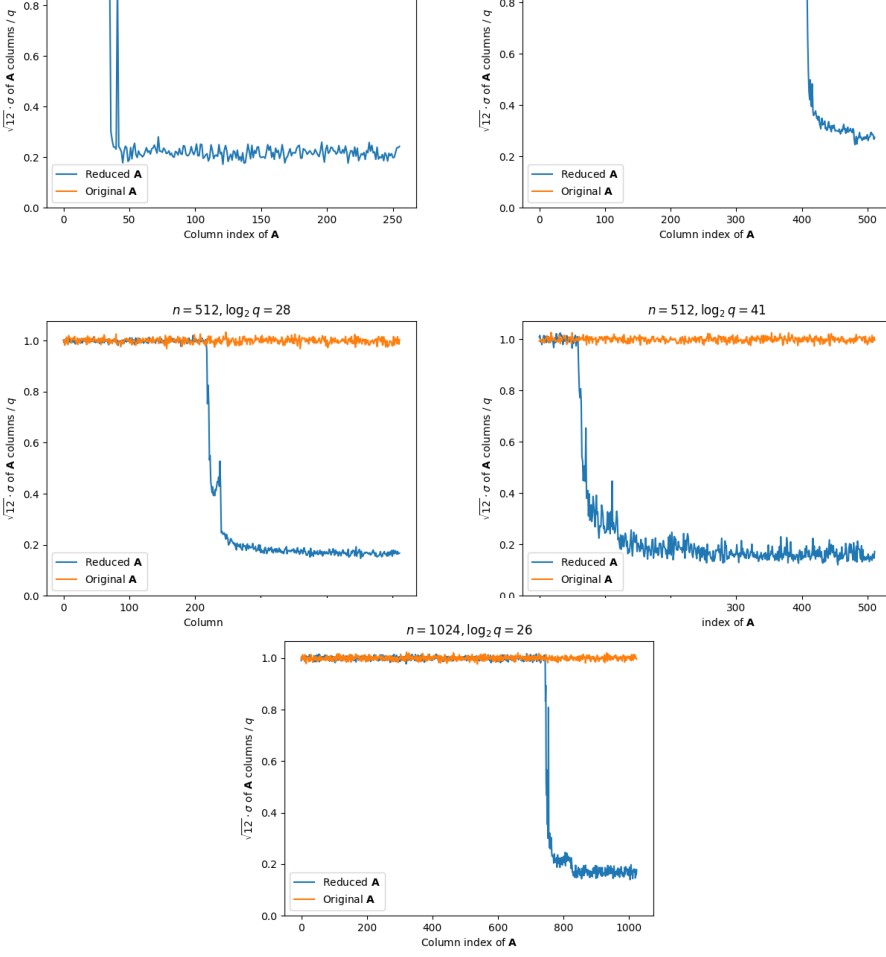

Figure 2: **Cliff shape in our five reduced datasets.** *Datasets that are more reduced have fewer columns of* **A** *with a normalized standard deviation of 1.0 (shorter cliff).*

## 4.2 Computational Cost

In Table 4, we report the number of CPU hours required to process one matrix in each setting. The reduction time depends on the $\tau$ threshold and the size of $n$ and $m$. There is also a tradeoff between BKZ block size and CPU hours and the reduction achieved. With experimentation, we find a block size that balances both. We see that the $n = 256$ dataset has the longest CPU hours per matrix, likely because it has the most aggressive reduction threshold $\tau$ relative to the problem difficulty.

Table 4: **Computational cost of generating datasets provided by this work**. *$n$ and $q$ are the LWE dimension and modulus, respectively. # hours/matrix is the average hours needed to reduce a matrix to the given $\tau$. Total CPU hours is hours to reduce a matrix on a single CPU times the number of matrices needed for 400M ($n = 256$) or 40M ($n = 512, 1024$) samples.*

| $n$ | $\log_2 q$ | # hours/matrix | Total CPU Hours |
|------|------------|----------------|-----------------|
| 256  | 20         | 44.31          | 40,486,047      |
| 512  | 12         | 7.34           | 349,090         |
| 512  | 28         | 16.88          | 664,971         |
| 512  | 41         | 15.2           | 647,721         |
| 1024 | 26         | 20.94          | 318,958         |

### 4.3 Benchmark Results

We run the SALSA and Cool and Cruel attacks—the two existing AI attacks on LWE—to establish baseline performance on the datasets. Both attacks are described briefly in §2, but we refer the interested reader to [36, 23, 34] for additional details. Here we give an overview of our implementation of each attack.

**SALSA.** We follow the setup of [36], using their open source code[3]. We train encoder-only transformers with 4 layers, embedding dimension 512, and 8 attention heads. We leverage the Adam [18] optimizer with a target learning rate of $l = 1e - 4$, 8000 warmup steps, and weight decay of $1e - 3$. We adopt the angular embedding technique of [34], which translates input elements of $\mathbf{a} \in \mathbb{Z}_\mathbf{q}^\mathbf{n}$ into coordinates on the unit circle. Finally, we use a training batch size of 64 and run the distinguisher algorithm (again, adopted from [36]) on 128 samples. We train the model on a single NVIDIA Titan X GPU for up to 3 days or 1000 epochs.

**Cool and Cruel.** Again, we follow the setup and use the open source code of [36], which improves upon the initial Cool and Cruel attack of [23]. In particular, we use the linear regression approach innovated by [36] to recover the cool bits, after using the simple brute force approach to recover cruel bits. We use 100, 000 data points for the brute force recovery and the rest for linear regression on the cool bits. We run each attack on a single NVIDIA Titan X GPU.

**Results.** Drawing on prior results [36], we generate binary secrets with varying Hamming weight and 3 or 4 cruel bits for each dataset, since these are known to be in the recoverable range for existing AI attacks. Then we run both attacks on these secrets. Out of all these experiments, we report the highest Hamming weight secret recovered for each setting in Table 5. These experiments sanity check the quality of the data by showing that sparse secrets can be recovered, and they establish baseline attack performance.

Table 5: **Benchmark results of ML attacks on the datasets provided by this work**. *n and q are the LWE dimension and modulus, respectively. h is the hamming weight of the secret (the number of non-zero elements in the secret vector of size n).*

| $n$ | $\log_2 q$ | max $h$ recovered (binary) | | max $h$ recovered (ternary) | |
|---|---|---|---|---|---|
| | | SALSA | CC | SALSA | CC |
| 256 | 20 | **33** | 22 | 20 | **22** |
| 512 | 12 | **6** | **6** | 4 | **7** |
| 512 | 28 | 9 | **12** | 11 | **13** |
| 512 | 41 | **63** | 60 | **45** | 37 |
| 1024 | 26 | 5 | **6** | 4 | **5** |

## 5 Related Work

Numerous works have proposed benchmarks for AI performance on mathematical reasoning tasks, which are somewhat related to our work. These include GM8SK [12], a grade school math reasoning benchmark; MATH [17], which includes high-school level math problems; and FrontierMath [15], an even more complex dataset of mathematical reasoning problems. These—and many other—math datasets are used to benchmark mathematical reasoning capabilities of large language models.

TAPAS is distinct from these generic math datasets in several ways. Instead of assessing generic mathematical capabilities, TAPAS datasets determine if models have learned specific mathematical capabilities—namely modular addition and multiplication—necessary for solving LWE. As a result of understanding these math concepts, models trained on our datasets also solve a useful problem: recovering LWE secrets. The dual nature of our datasets sets them apart.

Prior work on AI attacks for LWE [36, 20] released some reduced datasets for training AI models on the LWE problem. However, these datasets only have 4M examples, limiting attack performance. We release much larger datasets to aid investigation of how data scaling affects the model's reasoning

---

[3]https://github.com/facebookresearch/LWE-benchmarking

capabilities. We also provide data for different parameter settings compared to prior work to enable exploration into how attack performance changes with the LWE parameters.

# 6    Discussion and Conclusion

This work presents TAPAS, a new collection of datasets to enable further study of AI-powered attacks on the Learning with Errors problem. These datasets are 10x bigger than those provided by prior work, containing at least 40 million reduced examples per LWE setting. For one setting ($n = 256$, $\log_2 q = 20$), we provide 400 million reduced LWE examples. Our benchmark experiments provide a baseline for current state-of-the-art AI-powered attack performance on these datasets, to serve as a starting point for community-wide efforts to improve on these attacks.

We believe these datasets hold immense value for the AI community, beyond the obvious benefits to those assessing the security of post-quantum cryptosystems. They provide an easy on-ramp for AI practitioners to experiment with a new problem domain—using AI models to recover secrets from LWE problems. To outperform current state-of-the-art attacks, model trainers must address complex challenges well-known in the AI community, such as models' difficulty in solving modular arithmetic. Creative solutions to these challenges will enable progress on LWE cryptanalysis specifically, while also advancing the broader field of machine learning.

**Limitations.** Prior work observed a correlation between the amount by which LWE data is reduced and the complexity (as measured by Hamming weight) of recoverable secrets—more reduced data enables recovery of higher Hamming weight secrets [21]. While the datasets provided in this work are much *larger* and more *diverse* (e.g. more parameter settings) than those previously made available, they are still on par with the reduction quality of prior datasets. Improvements in reduction quality require innovation in lattice reduction techniques, a line of inquiry outside the scope of this paper. Future work could explore whether simply training AI models on more data could mediate the effect of truncated reduction quality, as more learnable patterns may be discerned in larger amounts of data. Furthermore, our datasets could also enable the study of scaling laws for how the amount of LWE data used in AI training affects secret recovery. In addition, generating these datasets required significant computational resources. To reduce this computational burden, future work can explore: data augmentation, synthetic data generation, and efficiency improvements. By making dataset generation more efficient, we can create larger training datasets that enhance AI attacks.

# 7    Future Work Leveraging Our Datasets

We hope that the datasets provided in this paper catalyze future AI-powered analysis of LWE security. Here are some suggestions for how AI researchers could leverage our datasets to advance the science of AI-enabled cryptanalysis of LWE (and beyond).

Current work on AI cryptanalysis of LWE  [36] has only explored the use of transformers in attacking LWE. Future work should consider other model architectures and/or training paradigms (such as reinforcement learning) to improve attacks. Perhaps a fusion approach, in which the pattern mining capabilities of AI models support traditional cryptanalysis techniques, would unlock progress.

Additionally, by treating LWE samples as data rather than as algebraic entities, we can explore statistical correlations in the data and other interesting properties that might yield cryptanalytic insights. [23] already demonstrated this potential by crafting a novel attack based on the *shape* of reduced data that had not previously been observed. Therefore, we encourage other statistical investigation of large-scale LWE datasets to surface other statistical signals.

In this work, we provide much larger training sets than previously reported. This should allow researchers to use larger models, train them for longer, and hopefully compute scaling laws for LWE attacks. Establishing scalings has proven beneficial in many applications of AI.

Publishing reduction matrices (together with the reduced LWE samples allows researchers to work on different secrets. This might allow models to be trained on some secrets and fine-tuned on others, a dramatic improvement to the efficiency of the current SALSA attack (which otherwise has to be rerun, preprocessing included, for every secret).

The reformulation of LWE as a pure arithmetic task (learning modular addition) will help integrate new paradigms from AI for Math, such as results on grokking [16] or modular arithmetic [30].

Finally, existing AI attacks formulate the problem of recovering the LWE secret in the same way: train a model $\mathcal{M}$ to predict $b$ given $\mathbf{a}$ with a fixed secret. However, this is not the only way the problem could be formalized. Researchers could use our datasets to investigate other cryptanalytic tasks of interest, like decision-LWE or distinguishability, by training models to distinguish between the provided (A, b) samples and random samples. Models could also be trained on many different LWE problems with different secrets, or perhaps trained on $(\mathbf{a}, b)$ pairs and asked to predict the secret $\mathbf{s}$. Such reformulations of the problem have not been explored and could possibly yield better generalization.

## Acknowledgments and Disclosure of Funding

The authors would like to thank Mohamed Malhou for his valuable feedback and insights.

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
