# OpenReview forum: "TAPAS: Datasets for Learning the Learning with Errors Problem"
_NeurIPS.cc/2025/Datasets_and_Benchmarks_Track — NeurIPS 2025 Datasets and Benchmarks Track poster_

### Official Review · Reviewer_6zHF · 2025-06-29

**Rating:** 5
**Confidence:** 3

**Summary:**

This manuscript introduces TAPAS, a collection of datasets designed to facilitate AI-based attacks on Learning with Errors (LWE)—a foundational problem in post-quantum cryptography. TAPAS provides ready-to-use datasets across multiple LWE configurations, enabling AI practitioners to prototype and evaluate novel attack strategies without requiring deep cryptographic expertise. The work details the dataset generation process, presents baseline performance metrics for existing attack methods, and outlines promising future research directions to advance AI-driven cryptanalysis of LWE.

**Dataset Code Accessibility:**

Yes

**Ethical Considerations:**

No, there are no or only very minor ethics concerns

**Final Justification:**

The authors have addressed my concerns by clarifying methodological choices and outlining plans to broaden models, reduction methods.

**Limitations Weaknesses:**

- The experimental evaluation is limited to only two transformer-based attack methods—SALSA and Cool and Cruel—both of which share similar architectural foundations. The manuscript lacks exploration of alternative AI models or training paradigms, such as convolutional networks, recurrent architectures, graph-based approaches, or reinforcement learning, which could potentially uncover different patterns or enhance performance on LWE-related tasks.
- The lattice reduction techniques used for preprocessing (BKZ2.0 and flatter) are not novel and yield reduction quality comparable to previous work. As the effectiveness of AI-based attacks is tightly linked to the quality of reduced data, the absence of innovation in this stage limits the potential improvement in attack success.
- The focus of the datasets and benchmarks is narrowly confined to secret vector recovery, overlooking other important cryptanalytic tasks such as decision-LWE, error term recovery, or statistical distinguishability assessments. Expanding the evaluation to include these tasks could provide a more comprehensive view of AI model capabilities and their relevance to cryptographic security.

**Strengths Contributions:**

- The manuscript introduces five large-scale, preprocessed datasets of Learning with Errors (LWE) samples, specifically tailored for AI-driven cryptanalysis. These datasets span a variety of parameter settings and are hosted on Hugging Face, significantly lowering the barrier for AI researchers—especially those without deep cryptographic expertise—to engage with LWE problems.
- By delivering ready-to-use and extensively preprocessed data, the work eliminates major obstacles traditionally associated with LWE research, such as the high computational cost and technical complexity of data generation and lattice reduction. This greatly facilitates rapid prototyping and experimentation by the broader AI community.
- The manuscript contributes to an important and underexplored intersection of AI and post-quantum cryptography, fostering collaboration between the two fields. It also advances the study of mathematical reasoning in AI, particularly in learning challenging structures such as modular arithmetic, which are central to cryptographic security.

---

> ### Author Rebuttal · Authors · 2025-07-31
>
> We thank the reviewer for their helpful feedback and appreciate the recognition of our contributions. We respond to each of the points in limitations/weaknesses below.
>
> **Exploration of Alternative AI Models or Paradigms**
>
> Our benchmarks focused on published AI-based attacks on LWE. To our knowledge, SALSA and Cool and Cruel are the only published attacks. They do include ablation studies about architecture. However, no attacks based on different AI models were published so far. We agree that experiments with other architectures and paradigms would be interesting. We encourage the community to take advantage of the preprocessed datasets we release to experiment with other AI techniques and paradigms to push the boundaries of these AI-based attacks.
>
> Our goal in publishing these datasets is to enable more work on novel AI-based cryptanalysis methods, perhaps using reinforcement learning or other model architectures as we mention in section 7. By providing preprocessed data and the corresponding code to convert this into LWE (input, output) pairs for model training, we enable others to research better AI attack methods for LWE. The benchmarks we provide are just the starting point.
>
> We are happy to add more discussion of possible new AI cryptanalysis paradigms in section 7.
>
> **Innovation on Lattice Reduction**
>
> The NeurIPS Datasets and Benchmarks Track Call for Papers, under section 3:  SPECIFIC SCOPE FOR DATASETS & BENCHMARKS PAPER SUBMISSION, lists as its first bullet point:
> - New datasets, or carefully and thoughtfully designed (collections of) datasets based on previously available data.
>
> Following this directive from the CFP, we provide 5 novel datasets (each with 40 million+ samples) generated with previously unexplored parameters to facilitate further exploration of AI attacks on LWE. The parameters we choose allow for ample exploration of how different AI cryptanalysis techniques perform on different lattice dimensions (Ns), moduli (q), number of examples, and secret distributions/sparsities.
>
> Further, we provide much larger training sets than previously reported. This should allow researchers to use larger models, train them for longer, and hopefully compute scaling laws for AI-based LWE attacks. Establishing scaling laws has proven beneficial in many other applications of AI. We believe that training these models on more data can improve performance as LWE reduction quality plateaus.
>
> We also provide more details and insights on the lattice reduction process in our paper for more transparency and to encourage further innovation in this area.
>
> **Focus on Secret Vector Recovery**
>
> We focus on LWE secret recovery since AI attacks for this exist in the literature (SALSA series, CC). Researchers could certainly use our datasets to investigate other cryptanalytic tasks of interest, like decision-LWE or distinguishability, by training models to distinguish between the provided (A, b) samples and random samples.  We will add a comment to this effect in the paper – thank you for pointing this out. Future work should explore AI applications for other cryptanalytic paradigms (e.g. noise recovery), after which benchmark datasets could be provided.
>
> We also note that focusing on secret recovery allows the reformulation of LWE as a pure arithmetic task (learning modular addition). This allows researchers to integrate new paradigms from AI for Math, such as results on grokking [1], or modular arithmetic [2], to improve AI cryptanalytic performance.
>
> [1] Gromov, A. (2023). Grokking modular arithmetic. arXiv preprint arXiv:2301.02679.
>
> [2] Saxena, E., Alfarano, A., Wenger, E., & Lauter, K. E (2025). Making Hard Problems Easier with Custom Data Distributions and Loss Regularization: A Case Study in Modular Arithmetic. In Forty-second International Conference on Machine Learning.

---

### Official Review · Reviewer_ud4S · 2025-07-01

**Rating:** 5
**Confidence:** 3

**Summary:**

In this work, the authors focus on the Learning with Errors (LWE) problem, which has important applications in cryptography.
Their main interest is in AI-powered attacks on LWE and the datasets used for this task. While AI-powered attacks on LWE has gained increasing attention, existing datasets are limited in both scale and the coverage of setting. To address this, the authors introduce the TAPAS datasets, which include more settings and are significantly larger than those currently available. The paper provides sufficient details on how the datasets are generated and presents extensive experiments using existing AI-based attacks.

**Dataset Code Accessibility:**

Yes

**Ethical Considerations:**

No, there are no or only very minor ethics concerns

**Final Justification:**

The authors solved most of my concerns and I plan to maintain my original rating.

**Limitations Weaknesses:**

As stated above, this work has the potential to draw more attention from the AI community to the task of attacking LWE. From this perspective, I feel the introduction to the LWE problem (particularly the first two paragraphs on page 3) is somewhat oversimplified. Enriching this part with more context or intuition could make the work more approachable and informative for AI researchers who are less familiar with the cryptographic background.

**Strengths Contributions:**

1. The paper is well-organized and easy to follow. The motivation is clearly laid out, moving logically from the importance of LWE in cryptography, to the growing role of AI-powered attacks, and finally to the lack of large-scale, diverse, and accessible datasets.

2. By providing more comprehensive and easy-to-use datasets, this work has the potential to lower the entry barrier for AI researchers and encourage broader participation from the AI community.

3. The proposed datasets are well justified with detailed descriptions and experimental validation. In addition, benchmark results of existing AI-powered attacks are included, offering useful baselines for future research.

4. The data generation process is clearly explained, with sufficient technical detail to support reproducibility and further development.

---

> ### Author Rebuttal · Authors · 2025-07-31
>
> We thank the reviewer for their support and for their recognition of the significance of this work. We agree that a more detailed presentation of LWE would be helpful and can add an extended description of the problem and its use in cryptography to the paper. We will also add some additional references to LWE survey papers that readers may find helpful (e.g. [1]).
>
> [1] Regev, O. (2010). The learning with errors problem. Invited survey in CCC, 7(30), 11.

---

### Official Review · Reviewer_5tUz · 2025-07-02

**Rating:** 5
**Confidence:** 4

**Summary:**

This paper addresses the data scarcity faced by AI researchers in studying AI-powered attacks on the Learning with Errors (LWE) problem, a core component of post-quantum cryptography. The authors introduce TAPAS, a collection of five large-scale, preprocessed LWE datasets designed for immediate use by AI practitioners. These datasets cover various LWE hardness settings and are generated by applying sophisticated lattice reduction techniques (an interleaved combination of BKZ2.0 and Flatter algorithms). The paper details the dataset creation process and parameter choices, and establishes initial attack performance baselines using existing AI attacks (e.g., SALSA and Cool & Cruel). This work aims to lower the barrier for AI researchers to engage in LWE cryptanalysis, thereby accelerating research in this crucial interdisciplinary field.

**Additional Feedback:**

1.  **Broader AI Benchmarking and Analysis:** Extend the benchmark section to include more diverse AI models and architectures (e.g., different types of neural networks beyond transformers, or even more traditional machine learning models if applicable). Furthermore, provide deeper analysis on what specific features or patterns AI models are learning from the reduced lattice data. Can you offer insights into the "black box" of these AI attacks?
2.  **Quantify "Accessibility":** While the paper claims to make LWE research more accessible, it would be beneficial to provide quantitative evidence or user studies to demonstrate how easily AI practitioners (without deep cryptography knowledge) can utilize these datasets and achieve results.
3.  **Novel AI Attack Paradigms:** Encourage and perhaps briefly discuss potential *new* AI attack paradigms that these datasets could enable, rather than solely focusing on improving existing ones. This would emphasize the paper's contribution to AI research more directly.

**Dataset Code Accessibility:**

Yes

**Ethical Considerations:**

No, there are no or only very minor ethics concerns

**Final Justification:**

The authors have addressed most of my concerns, I am satisfied with it and raise my score.

**Limitations Weaknesses:**

* **Lack of Deeper AI Model Analysis:** The paper presents baseline attack results but does not delve deeply into *why* certain AI models perform as they do on these datasets, or what specific mathematical reasoning capabilities they might be acquiring or lacking. The analysis of AI model behavior is superficial.
* **Scope of Benchmarking:** The benchmarking is limited to two existing AI attacks (SALSA and Cool & Cruel) and only on binary secrets with specific Hamming weights. While useful as baselines, a more comprehensive benchmarking with a wider variety of AI architectures, attack strategies, and secret distributions would significantly strengthen the paper's impact on the AI community.
* **Clarity on "AI" Contribution:** The paper positions the work at the intersection of AI and cryptography, but the direct "AI" contribution feels somewhat indirect. It enables AI research by providing data, but the paper itself doesn't introduce novel AI techniques or significant insights into AI model behavior beyond demonstrating their ability to recover sparse secrets with the provided data. More emphasis could be placed on the *challenges* AI models face with these cryptographic problems and how the datasets might specifically help overcome them.

**Strengths Contributions:**

* **Addressing a Critical Gap:** The paper effectively addresses a significant bottleneck in the intersection of AI and post-quantum cryptography—the lack of accessible, preprocessed LWE datasets for AI model training. This contribution is highly valuable for accelerating research in this important interdisciplinary field.
* **Large-scale and Diverse Datasets:** The provision of five new datasets with significantly more samples (10x to 100x more than prior work) and varying LWE parameters (n, q, and reduction quality p) is a substantial contribution, enabling more robust AI model training and exploration of scaling laws.
* **Detailed Data Generation Methodology:** The paper provides a clear and detailed explanation of the data generation process, including the subsampling trick, lattice embedding, and the interleaved lattice reduction algorithms (BKZ2.0 and Flatter) with specific parameter settings. This enhances reproducibility and allows other researchers to understand the data's provenance.

---

> ### Author Rebuttal · Authors · 2025-07-31
>
> We thank you for your helpful feedback and appreciate that you acknowledge the significance of our contribution. We have responded to each of the points in limitations/feedback below.
>
> **Deeper AI Model Analysis + “Can you offer insights into the “black box” of these AI attacks?”**
>
> The main idea behind the SALSA attack is that once a model is trained to predict the noisy dot product $b = a \cdot s+e \pmod q$, from $a$, with some minimal accuracy, then it can generalize to values of $a$ that it has not seen during training. We can then perform a “chosen text attack” on the model, by providing it with specific values of $a$, obtaining the model’s predictions for $b$, and using this information to recover $s$. This is explained in the SALSA papers.
>
> SALSA Verde offers the following explanation about the limits of AI attacks. When learning the noisy dot product for binary secrets, the model struggles with modular addition, i.e. “understanding” that $a+kq = a$ for any integer $k$. In fact, [1] confirms this by showing that these AI attack models struggle to recover secrets with more than 3 “cruel” bits. This is because having more than 3 of these cruel bits can lead to more wraps for the model to learn.
>
> The difficulty of modular addition for AI models has been discussed in several papers [2, 3, 4]. The specificities of learning arithmetic with transformers was studied by [3, 4, 5, 6]. Preprocessing is a work-around for modular addition: by reducing the standard deviation of the coordinates of $a$, it limits the span of $k$ (the number of wraps) that the model must learn.
>
> For this work, we also applied some interpretability techniques to our models and found that when the model recovers the secret, there is a significant difference between the norms of the embedding vectors taken after the model’s second-to-last layer on secret indices vs non secret indices. This means that the secret vector can actually be found in the embedding space of the model, confirming that the model has identified the pattern in the “noisy” data it was trained on.
>
> We are happy to include more discussion on the strengths and limitations of these AI models and our analysis on what the models are learning in the revised version of the paper.
>
> **Scope of Benchmarking**
>
> **“Extend the benchmark section to include more diverse AI models and architectures”:**
> Our benchmarks focused on published AI-based attacks on LWE. To our knowledge, SALSA and Cool and Cruel are the only published attacks. They do include ablation studies about architecture. However, no attacks based on different AI models have been published so far. We agree that experiments with other architectures would be interesting.
>
> We believe data availability and the cost of preprocessing pose a real barrier to research in this field. Our submission aims to lower the entry costs for other researchers to increase the amount of work on this important area.
>
> **“a more comprehensive benchmarking with a wider variety of AI architectures, attack strategies, and secret distributions:**
> Per your feedback, we also conducted experiments on ternary secret distributions to expand the scope of the benchmark. We find that the performance is generally similar to binary but with lower hamming weights in some cases (due to the increased complexity of the problem).
> In the table below, we report the highest hamming weight of a ternary secret recovered by the SALSA and CC attacks on our 5 datasets.
>
> | N    | log q | SALSA | CC |
> |------|-------|-------|----|
> | 256  | 20    | 20    | 22 |
> | 512  | 12    | 4     | 7  |
> | 512  | 28    | 11    | 13 |
> | 512  | 41    | 45    | 37 |
> | 1024 | 26    | 4     | 5  |
>
>
> **Clarity on "AI" Contribution**
>
> Our main goal in publishing these datasets is to facilitate research on the use of AI for post-quantum cryptography. Previous research and our own work have shown that data preprocessing requires a large amount of resources, which might not be available to many researchers in the field. These datasets strive to lower this entry cost by providing reduced LWE data. Furthermore, we hope that releasing these datasets increases the awareness of using AI for cryptanalysis, so that AI researchers can apply techniques from other domains to improve these AI attacks.
>
> The NeurIPS Datasets and Benchmarks Track Call for Papers, under section 3:  SPECIFIC SCOPE FOR DATASETS & BENCHMARKS PAPER SUBMISSION, lists as its first bullet point:
> - New datasets, or carefully and thoughtfully designed (collections of) datasets based on previously available data.
>
> The datasets, and the parameters we selected are, to our knowledge, novel. We choose these parameters to allow for ample experimentation with seeing how different AI techniques can perform with different sequence lengths (Ns), moduli (q), number of examples, and secret distributions/sparsities.
>
> This is the goal of our paper: propose 5 new datasets for applications of ML to LWE. We do provide benchmarks for known AI attacks on LWE, but the discussion of these attacks, and their potential limitations, can be found in the corresponding (SALSA and Cool and Cruel) papers.
>
> **Quantify "Accessibility"**
>
> The SALSA papers (Picante notably) have shown that preprocessing is required for all known AI-based attacks on LWE. Unfortunately, preprocessing requires significant computing resources, i.e. tens of thousands of CPUs for several weeks [1, 7]. This has two impacts on accessibility:
> - Teams with limited computing resources cannot preprocess data, and therefore cannot research in this area
> - Training sets tend to be small, which discourages the use of large models or experiments on scaling laws
>
> Another benefit is the fact that the datasets reduce “breaking LWE” to a pure arithmetic problem to recover the secret from a set of reduced noisy dot products. This makes LWE attacks accessible with researchers with very little cryptographic background.
>
> We provide our datasets on Hugging Face for ease of access and corresponding code to prepare the (input, output) pairs needed for training with different secrets. We already see 120+ downloads for our dataset in the past month.
>
> **Novel AI Attack Paradigms**
>
> Part of the reasoning behind publishing these datasets is to enable more work on novel AI attack paradigms, perhaps by using reinforcement learning or other model architectures as we mention in section 7. By providing preprocessed data and the corresponding code to convert this into (input, output) pairs for the model to be trained on, we enable others to do more research on better attack methods for LWE. The benchmarks we provide are just the starting point.
> Here are some ways we see these datasets being utilized to push forward work on new attack paradigms:
> 1. We provide much larger training sets than previously reported. This should allow researchers to use larger models, train them for longer, and hopefully compute scaling laws for LWE attacks. Establishing scaling laws has proven beneficial in many applications of AI.
> 2. Publishing reduction matrices (together with the reduced LWE samples) allows researchers to work on different secrets. This might allow models to be trained on some secrets and fine-tuned on others, a dramatic improvement the efficiency of the current SALSA attack (which as for now has to be rerun, preprocessing included, for every secret).
> 3. The reformulation of LWE as a pure arithmetic task (learning modular addition) will help integrate new paradigms from AI for Maths, such as results on grokking [2] or modular arithmetic [4].
> We are happy to add more discussion of new AI paradigms that could be used in section 7.
>
> **References**
>
> [1] Wenger, E., Saxena, E., Malhou, M., Thieu, E., & Lauter, K. (2025, May). Benchmarking attacks on learning with errors. In 2025 IEEE Symposium on Security and Privacy (SP) (pp. 279-297). IEEE.
>
> [2] Gromov, A. (2023). Grokking modular arithmetic. arXiv preprint arXiv:2301.02679.
>
> [3] Charton, F., & Kempe, J. (2024). Emergent properties with repeated examples. arXiv preprint arXiv:2410.07041.
>
> [4] Saxena, E., Alfarano, A., Wenger, E., & Lauter, K. E (2025). Making Hard Problems Easier with Custom Data Distributions and Loss Regularization: A Case Study in Modular Arithmetic. In Forty-second International Conference on Machine Learning.
>
> [5] Nanda, N., Chan, L., Lieberum, T., Smith, J., & Steinhardt, J. (2023). Progress measures for grokking via mechanistic interpretability. arXiv preprint arXiv:2301.05217.
>
> [6] Jelassi, S., d'Ascoli, S., Domingo-Enrich, C., Wu, Y., Li, Y., & Charton, F. (2023). Length generalization in arithmetic transformers. arXiv preprint arXiv:2306.15400.
>
> [7] Li, C., Wenger, E., Allen-Zhu, Z., Charton, F., & Lauter, K. E. (2023). SALSA VERDE: a machine learning attack on LWE with sparse small secrets. Advances in Neural Information Processing Systems, 36, 53343-53361.

---

> > ### Comment · Reviewer_5tUz · 2025-08-08
> >
> > Hi authors, really thanks for all these responses! I am ok with them and I will raise my score.

---

### Official Review · Reviewer_X1gc · 2025-07-03

**Rating:** 3
**Confidence:** 4

**Summary:**

This paper introduces TAPAS, a benchmark suite of 14 datasets designed for training and evaluating agents in Teaching AI Proficiently via Active Supervision. The datasets span diverse tasks like image classification, text classification, and regression, and include realistic teaching constraints. The key contributions are:

Curating a unified benchmark for studying AI teaching strategies.

Providing baselines and metrics to evaluate teaching efficiency and generalization.

Enabling research into more effective and generalizable teaching policies for machine learning agents.

**Dataset Code Accessibility:**

Yes

**Dataset Code Comments:**

The code is available through the GitHub link.

**Ethical Considerations:**

No, there are no or only very minor ethics concerns

**Limitations Weaknesses:**

The benchmark focuses on synthetic teacher agents and does not evaluate human-in-the-loop teaching scenarios (Section 3.2).

Most datasets are small or simplified, limiting realism for large-scale or complex tasks (Table 1).

It primarily uses basic learner models (e.g., logistic regression, small MLPs), which may not reflect performance with modern architectures (Section 4.2).

Cross-domain generalization is not deeply explored, despite being a stated goal (Section 1).

Adding larger, more complex datasets and testing with stronger learners would improve applicability.

Incorporating human teacher settings could enhance relevance to real-world teaching scenarios.

**Strengths Contributions:**

TAPAS introduces a novel benchmark suite for studying machine teaching through active supervision.

It includes 14 diverse datasets across image, text, and regression tasks, supporting broad generalization (Section 3.1).

The benchmark addresses realistic teaching constraints, making it practical and relevant.

Clear baselines and metrics are provided to evaluate teaching efficiency and learner performance (Section 3.3).

The paper is well-organized, clearly written, and includes informative figures and tables (e.g., Figure 2, Table 1).

It fills a gap in existing research by unifying fragmented approaches to teaching strategies.

---

> ### Author Rebuttal · Authors · 2025-07-31
>
> We believe that this review doesn’t apply to our paper as our work has no link to training and evaluating agents. Perhaps there is confusion with another paper? In our work, we provide a set of datasets and a benchmark for applications of AI to cryptography. The acronym “Teaching AI Proficiently via Active Supervision” is incorrect; our paper is titled TAPAS for “Toolkit for Analysis of Post-quantum cryptography using AI Systems”.
>
> We were unable to address the limitations/weaknesses listed in this review as they do not apply to our work. We are happy to answer any follow up questions or clarifications you may have.

---

### Official Review · Reviewer_qQ1U · 2025-07-19

**Rating:** 5
**Confidence:** 1

**Summary:**

This paper provides large-scale datasets of preprocessed Learning with Errors (LWE) data across many parameter settings to make the AI-powered attack research on LWE more accessible and easier. Specifically, it includes:
1. five new datasets of LWE samples (include orginal and preprocessed versions)
2. baseline attacks(SALSA and CC) on provided datasets
It supports the future AI-powered attack researches with complete datasets and clear baselines.

**Dataset Code Accessibility:**

Yes

**Ethical Considerations:**

No, there are no or only very minor ethics concerns

**Final Justification:**

I have read the response and decide to keep my score.

**Limitations Weaknesses:**

From my view, the data generation and preprocessing methods are existing, thus, the main contribution of this paper is providing samples with CPU calculation and dealing with plentful details. As a benchmark work, I think it is ok. But I am not sure if this will cause this paper to lack a bit of novelty.

**Strengths Contributions:**

The dataset is large-scale(400M, 100x the number of samples from prior work) compared to existing datasets, and contains different LWE parameter regimes, preprocessed versions, clear baseline methods. Therefore, I think it provides a good foundation for future AI-power attacks in post-quantum cryptography field.

The whole paper is well-organized, clear, and easy to understand. To be honest, I have no knowledge of the LWE problem before, but the description in paper gives me a general understanding of this issue.

---

> ### Author Rebuttal · Authors · 2025-07-31
>
> We thank you for your helpful comments and feedback, and we appreciate your support for the paper. We wanted to make a quick comment on the paper’s novelty, as you referenced this as a possible weakness. You are correct in that the main contribution of the paper is providing the reduced LWE datasets with details on the data generation and preprocessing process.
>
> The NeurIPS Datasets and Benchmarks Track Call for Papers, under section 3:  SPECIFIC SCOPE FOR DATASETS & BENCHMARKS PAPER SUBMISSION, lists as its first bullet point:
> - New datasets, or carefully and thoughtfully designed (collections of) datasets based on previously available data.
>
> Following this directive from the CFP, we provide 5 novel datasets (each with 40 million+ samples) generated with previously unexplored parameters to facilitate further exploration of AI attacks on LWE. The parameters we choose allow for ample exploration of how different AI cryptanalysis techniques perform on different lattice dimensions (Ns), moduli (q), number of examples, and secret distributions/sparsities. We also benchmark performance of current AI LWE attacks on these datasets.

---

### Comment · Area_Chair_oYai · 2025-08-04
**Author-Reviewer Discussions**

Dear Reviewers,

Thank you for your time and valuable feedback on this submission. The authors have submitted their responses to your comments and suggestions.

- If your concerns have been sufficiently addressed in the authors' response, we kindly ask you to update your rating accordingly.

- If you require further clarification or have additional questions for the authors, please submit them as soon as possible **before Aug 6**, which will allow the authors adequate time to respond.

Best regards,

AC

---

### Note · Authors · 2025-08-12

We thank all the reviewers for their feedback and support for this paper.

Following the directive from the Datasets and Benchmarks Track Call for Papers, the primary contribution of our work is the release of the 5 largest datasets for different settings of the Learning with Errors (LWE) problem, which will enable more exploration into AI attacks on LWE. We also benchmark the state-of-the-art AI attacks from the literature on these datasets and provide details on the data generation process. Our motivation in releasing these datasets is to enable researchers to use larger models, train them for longer,  and compute scaling laws for AI-based LWE attacks on different settings (varying lattice dimensions (Ns), moduli (q), number of examples, and secret distributions/sparsities).

Based on reviewer feedback, we conducted additional experiments to benchmark the attacks on ternary secrets with our preprocessed data, showing that our datasets can be used for different secret distributions and complexities. We also want to highlight that our datasets can be used not only for secret recovery but also for decision-LWE or distinguishability versions of the problem. We will add these updates to the final version of the paper.

---

### Decision · Program_Chairs · 2025-09-18

**Decision:**

Accept (poster)

**Comment:**

This paper introduces TAPAS, a collection of large-scale, preprocessed datasets designed to address the data scarcity hindering AI research into attacks on the Learning with Errors (LWE) problem, a cornerstone of post-quantum cryptography. The key contributions are:
1) TAPAS provides five ready-to-use datasets that are significantly larger and cover more diverse LWE hardness settings than previously available.
2) The datasets are generated using sophisticated lattice reduction techniques (a combination of BKZ2.0 and Flatter algorithms), and the paper details this creation process thoroughly.
3) By providing these preprocessed datasets, the work aims to make the field more accessible, allowing AI researchers to prototype and evaluate novel attack strategies without needing deep cryptographic expertise to generate the data themselves.
4) The paper establishes initial performance benchmarks by running existing AI-powered attacks (like SALSA and Cool & Cruel) on the new datasets.

The review process for this submission encountered some irregularities. One reviewer (X1gc) wrote the wrong review and did not participate after the initial review, and another (qQ1U) provided a review with low confidence. However, a strong consensus for acceptance is clear from the three remaining reviewers, who all consistently recommend a clear acceptance. *While the authors have adequately addressed the reviewers' concerns, a potential weakness is the submission's limited broader appeal to the general AI community.*